# Synergistic Protection of Retinal Ganglion Cells (RGCs) by SARM1 Inactivation with CNTF in a Rodent Model of Nonarteritic Anterior Ischemic Optic Neuropathy

**DOI:** 10.3390/cells13030202

**Published:** 2024-01-23

**Authors:** Yan Guo, Zara Mehrabian, Jeffrey Milbrandt, Aaron DiAntonio, Steven L. Bernstein

**Affiliations:** 1Departments of Ophthalmology and Visual Sciences, School of Medicine, University of Maryland, Baltimore, MD 21201, USA; yanguo@som.umaryland.edu (Y.G.); zmehrabyan@som.umaryland.edu (Z.M.); 2Department of Genetics, Washington University School of Medicine, St. Louis, MO 63110, USA; jmilbrandt@wustl.edu; 3Needleman Center for Neurometabolism and Axonal Therapeutics, St. Louis, MO 63110, USA; diantonio@wustl.edu; 4Department of Developmental Biology, Washington University School of Medicine, St. Louis, MO 63110, USA; 5Anatomy and Neurobiology, School of Medicine, University of Maryland, Baltimore, MD 21201, USA

**Keywords:** optic nerve, nonarteritic anterior ischemic optic neuropathy (NAION), sterile alpha and (toll/interleukin receptor (TIR)) motif-containing 1 (SARM1), rodent, ischemia, axonopathy, retinal ganglion cells, neuroprotection, ciliary neurotrophic factor, synergism

## Abstract

We evaluated whether inhibiting sterile alpha and (Toll/interleukin receptor (TIR)) motif-containing 1 (SARM1) activity protects retinal ganglion cells (RGCs) following ischemic axonopathy (rodent nonarteritic anterior ischemic optic neuropathy: rNAION) by itself and combined with ciliary neurotrophic factor (CNTF). Genetically modified SARM1(−) rats were rNAION-induced in one eye and compared against equivalently induced wild-type animals of the same background. Optic nerve (ON) diameters were quantified using optical coherence tomography (SD-OCT). RGCs were quantified 30 d post-induction using retinal stereology for Brn3a(+) nuclei. ON sections were analyzed by TEM and immunohistochemistry. SARM1(−)(−) and WT animals were then bilaterally sequentially rNAION-induced. One eye received intravitreal vehicle injection following induction; the contralateral side received CNTF and was analyzed 30 d post-induction. Inhibiting SARM1 activity suppressed axonal collapse following ischemic axonopathy. SARM1(−) animals significantly reduced RGC loss, compared with WT animals (49.4 ± 6.8% RGC loss in SARM1(−) vs. 63.6 ± 3.2% sem RGC loss in WT; Mann–Whitney one-tailed U-test, (*p* = 0.049)). IVT-CNTF treatment vs. IVT-vehicle in SARM1(−) animals further reduced RGC loss by 24% at 30 d post-induction, but CNTF did not, by itself, improve long-term RGC survival in WT animals compared with vehicle (Mann–Whitney one-tailed *t*-test; *p* = 0.033). While inhibiting SARM1 activity is itself neuroprotective, combining SARM1 inhibition and CNTF treatment generated a long-term, synergistic neuroprotective effect in ischemic neuropathy. Combinatorial treatments for NAION utilizing independent neuroprotective mechanisms may thus provide a greater effect than individual treatment modalities.

## 1. Introduction

Nonarteritic anterior ischemic optic neuropathy (NAION) is the most common cause of sudden optic nerve (ON)-related vision loss in individuals over the age of 50 [1,2]. NAION is an ischemic axonopathy at the most anterior portion of the ON, resulting in infarction of the retinal ganglion cell (RGC) axons. Causation is typically associated with a ‘disk at risk’ (small ON outlet from the eye) [3,4], but multiple factors can contribute to this condition [5,6]. NAION typically occurs from a compartment syndrome following vascular decompensation and loss of capillary support leading to optic nerve head ischemia [2,7,8]. The ischemic axonopathy typically results in sudden unilateral or sequential bilateral loss of vision that can range from complete blindness to a regional visual loss of varied severity [2,9].

There are currently no clinically effective treatments for NAION [10,11]. This in part results from a lack of understanding of the root causes, but an important contributing factor includes the prolonged window of time between disease onset and diagnosis, which typically involves 1 day or more [12,13]. Some individuals can be unaware of the visual loss until occlusion of the noninvolved eye shows a visual defect. 

A rodent NAION model (rNAION) has been developed that exhibits many of the features of the clinical NAION [14,15]. This model exhibits many common mechanisms with the disease, including optic nerve head (ONH) edema, compartment syndrome followed by axonal collapse, and isolated RGC stress and death through apoptosis. Focal humoral inflammatory activation is followed by systemic macrophage invasion of the affected infarct. Later, systemic macrophages are replaced with activated microglia. The rNAION model has also been useful in evaluating potential treatments for NAION, and a number of approaches have been shown to be neuroprotective in the earliest stages of the model, or as pre-treatments [16,17,18]. These include the following: edema suppression [19], Rho-kinase inhibitors [20], inflammatory suppression [21,22], growth factors such as ciliary neurotrophic factor (CNTF) [23], brain-derived neurotrophic factor (BDNF) and its ligands [24,25], and anti-apoptotic agents [26]. However, these individual factors’ neuroprotective effects are modest even when administered close to lesion onset, and BDNF and CNTF’s RGC-neuroprotective effects were only quantified at 15 d post-induction [23]. Thus, there is a need for more efficacious strategies for NAION treatment that are effective at extended treatment time windows, and exert long term protection. 

Axon loss is an early event in rNAION, and so therapeutic strategies to block pathological axon degeneration (Wallerian degeneration) may prove efficacious in diseases involving axonal damage [27,28,29]. In recent years, tremendous progress has been made in delineating the molecular mechanisms of Wallerian degeneration. SARM1 (sterile alpha and toll/interleukin receptor (TIR) motif-containing 1) is an inducible NAD+ cleaving enzyme that is the central executioner of pathological axon degeneration [30,31]. SARM1 is a metabolic sensor of the ratio of NAD+ to its precursor NMN [32,33] and is activated when the labile axonal NAD+ biosynthetic enzyme NMNAT2 is lost following disruptions to axonal transport [34]. Genetic deletion of SARM1 is profoundly protective in rodent models of traumatic nerve injury, traumatic brain injury, peripheral neuropathy, and chemotherapy-induced peripheral neuropathy [21,35,36,37], as well as slowly progressive axonopathies including CMT2A [38]. Additionally, SARM1 deletion enhances RGC survival in models of RGC stress and damage such as neuroinflammatory [39] and pressure models of glaucoma [35], although not in optic nerve crush or EAE [35,40]. Moreover, SARM1 is a druggable target [41], with both small molecule inhibitors and gene therapeutics under development [42,43,44], and so there is great interest in identifying diseases in which SARM1 contributes to neuropathology. Our first aim was to evaluate whether eliminating SARM1 activity can enhance long-term RGC survival in focal axonal ischemia, using the rat rNAION model in a recently developed rat SARM1(−) knockout line generated in an SD rat strain using Crispr-Cas9 technology [38].

Previous investigations have revealed that combinatorial therapies utilizing well-considered, independent, neuroprotective, or regenerative mechanisms can yield synergistic or additive effects that far exceed individual therapies [45,46]. A number of rNAION-associated neuroprotective treatments with independent mechanisms have already been identified, and even apparently ineffective individual approaches may function in a synergistic fashion when combined with other treatments. CNTF has been shown to delay RGC death in culture in vivo in a variety of ON damage models, including ON crush and models of glaucoma [47,48,49]. Our second aim was to evaluate the effects of combining SARM1(−)-delayed axonal degeneration with suppression of RGC somatic degeneration using CNTF. 

## 2. Materials and Methods

**1.** Animals: SARM1 knockout rats on a Sprague-Dawley (SD1) background were generated as previously described [38] and kept as homozygous stock. Animals were outbred with SD-wild-type (WT) animals every five generations. SD-WT animals were also rNAION-induced for use in comparing histological responses against SARM1 homozygotes. Because the SARM1(−)(−) strain is on an SD-albino background, and albino animals have reduced visual acuity compared with pigmented animals, we utilized direct anatomical analysis using RGC quantification and ON histology rather than functional tests. It should be noted that deleting SARM1 activity has also been shown to preserve spatial vision in toxic optic neuropathies in pigmented animals [50]. A total of 21 WT animals and 27 SARM1(−)(−) animals were used for all aspects of this study.

**2.** SARM1 genetic analysis: SARM1 knockout strain difference was confirmed using DNA templates prepared from tissue from ear clips and fragment amplification using the polymerase chain reaction (PCR). We used SARM1 WT primers’ amplified DNA isolated from ear clip tissue and compared it against WT and homozygous standards. The forward primer was TCGGCCTAGGCGTGATCTTG, and the reverse primer was GGCTTGTGTCACTGGCATCC. Predicted fragment sizes were 530 bp for WT and 420 bp for SARM1 homozygotes.

**3.** rNAION induction: rNAION was induced as previously described [14], except that a 532 nm diode laser (Oculite GLx; Iridex, Mountain View, CA) was used. The laser power was 51 ± 2 mW measured at the point of pupil entry by a pyroelectric detector-laser power meter (Coherent; Fremont, CA). The corneas of anesthetized animals were topically anesthetized with proparacaine, and the pupils were dilated with 1% tropicamide. A custom contact lens (Micro lens-R; Nissel and Cantor, UK) was applied. Animals were injected with sterile rose Bengal (2.5 mM in isotonic saline), followed 30 s later with 11 s laser exposure at 500 μM spot size. For initial experiments (direct comparison of rNAION responses in SARM1 homozygotes against SD-WT), only one eye of each animal was induced. For vehicle vs. CNTF treatment, animals of both strains were sequentially induced in both eyes one week apart. Because the rNAION procedure reduces visual function but does not blind the animal, the IACUC committee found that it was ethically appropriate to treat both eyes of the same animal in the bilateral vehicle/CNTF treatment experiment, and they reduced the overall number of animals required for analysis.

**4.** Design of CNTF comparative treatment effectiveness trial for WT and SARM1(−)(−) animals: We utilized a bilateral treatment approach to determine the comparative neuroprotective effectiveness of vehicle (PBS) and CNTF, since it reduces the inter-individual response difference between the two treatments and can help uncover the actual protective effect of drugs. We minimized the potential effect of induction and intravitreal injection stress on the second eye by using a 1-week washout interval between treatments and by using a placebo for the treatment of the first induced eye. Following rNAION induction of the first eye, animals were intravitreally injected unilaterally with 2 μL of vehicle (PBS, pH 7.4). The second eye of the same animal was induced one week after induction of the first eye and injected with 2 μL of 50 ng CNTF in PBS. Both WT (SD) and SARM1 animals were compared.

**5.** SD-OCT analysis of ONH edema: ONH edema was quantified two days post-induction in each eye by SD-OCT (Heidelberg), as initially described [51]. Briefly, the mean inner nuclear layer (INL–INL) distance separated by the ON at the widest point was measured two days post-induction for each eye. Animals with mean ONH diameters < 500 μm (threshold) were excluded from further study.

**6.** Transmission electron microscopy: Single animals (WT and SARM1(−)(−) were used for this part of the study since only the axonal structure changes at specific times were evaluated and axon numbers were not quantified. Animals were intracardially perfused with 4% paraformaldehyde in phosphate-buffered saline (PBS) pH 7.4, along with 0.5% glutaraldehyde to prevent myelin dissociation. Optic nerve tissues were post-fixed in a mixture of glutaraldehyde-paraformaldehyde (4-FIG) overnight prior to uranyl acetate staining and upon embedding. Sectioning was performed at 500 nm thickness and flattened on copper grids. Sections were examined using a Tecnai T12 transmission EM (FEI; Hillsboro, OR, USA) at 2100X and 4400X.

**7.** Immunohistology: Whole retinae were stained with goat anti Brn3a primary antibody (Santa Cruz: RRID AB_630987) and reacted with Cy3 fluorescent donkey anti-goat secondary antibody (Jackson Immunoresearch) and flat-mounted for RGC stereology. ON sections were reacted with mouse monoclonal antibody SMI312 (Abcam RRID_AB448151) for neurofilament identification and polyclonal rabbit anti-myelin basic protein (MBP: Abcam RRID_AB1841021). The immunohistology-based structure was examined using both fluorescent microscopy (Keyence BZX) and a confocal microscope (Leica).

**8.** Early axonal loss and demyelination analysis: Single animals (both WT and SARM1(−)(−) were used for this part of the analysis since only a gross comparison of relative axonal preservation and myelination differences between WT and SARM1(−)(−) nerves at early (5 d) times during ischemic conditions was desired. ON sections were reacted with SMI312 and anti-MBP antibodies (see immunohistology), and a relative number of intact, myelinated axons were quantified in the individual sections.

**9.** RGC loss analysis: Animals used in the WT–SARM1 initial comparison were unilaterally induced, and % RGC loss was calculated by comparing the ratio of Brn3a(+) nuclear counts from the uninduced eye against the induced eye. This yielded a direct comparison percentage of post-rNAION RGC loss for WT and SARM1 knockout animals.

RGC loss for animals used in the combinatorial experiment (SARM1 + CNTF) was determined 30 days after sequential induction and treatment of both eyes of each animal (WT and SARM1 knockout) 1 week apart. Animals were held for 30 days after the second eye induction and then euthanized. 

RGC quantification was performed for both experiments using randomized computer-driven stereology. Brn3a(+) immunohistochemical staining was performed on the whole retinae [52], and RGCs were counted from flat-mounted retinae using a Nikon E300 fluorescent microscope with a computer-driven, motorized stage. The RGC ratio (RGCs/total retinal area) ratio was calculated for each eye, and the treatment difference was calculated for each bilaterally induced animal. 

## 3. Results

### 3.1. Loss of SARM1 Activity Inhibits Early RGC Axonal Degeneration

The ischemic lesion induced by threshold rNAION is related to a compartment syndrome with a severe compromise of the smallest capillaries in the center of the nerve, with relative preservation of circulation in areas of the peripheral regions of the nerve. Thus, axonal damage is typically worst in the ischemic ON center, with less pronounced axonal loss peripherally and occasional preservation of axonal fields near the ON sheath. Single animals 5 d post-induction were used for this portion (Section 3.1 and Section 3.2) of the study. Normal ON axons in the periphery of the rNAION-induced ON have granular appearing axoplasm and closely adherent myelin sheaths (Figure 1A, WT). ON axons vary widely in their widths (Figure 1A). Ischemic WT RGC axons undergo collapse by five days post-rNAION (Figure 1C,D) (Figure 1C,D ON center; compare with Figure 1A: axons in the ON periphery). Early axonal damage in the central region is characterized by axonal swelling and the appearance of both dark bodies in the axoplasm (Figure 1C: white arrowheads), as well as areas of diffuse radiolucent edema (Figure 1D: double arrows). There is a mixed pattern of axonal damage in the mid-periphery, with some (typically smaller diameter) axons remaining undamaged and surrounded by other axons undergoing degeneration (Figure 1B: double arrows). SARM1(−)(−), rNAION-induced ONs show a similar pattern of normal-appearing axons peripherally (Figure 1E), but while there is some patchy swelling of myelin sheaths in some of the axons in the mid-peripheral region (Figure 1F; white arrowheads), the axoplasm itself appears intact. Axons in the center of the SARM1(−)(−) ON show myelin fragmentation (Figure 1G,H; arrows) and axoplasmic degradation in multiple axons (Figure 1G; asterisks), but the axoplasm of many other axons appears intact, and there are intact appearing smaller myelinated axons scattered throughout the field (Figure 1G,H; double arrowheads). This suggests that loss of SARM1 activity at least modestly increases axonal resistance to acute ischemic axonopathy in the early stages of the disorder.

### 3.2. SARM1 Loss Inhibits Early ON Axonal and Myelin Degeneration

SARM1(−) animals show delayed axonal degeneration and distal axon loss following ON crush or section [40], but suppressing SARM1 activity in these models does not prevent ultimate axonal loss. Five days post-rNAION induction, immunohistochemical analysis reveals considerable differences in WT vs. SARM1(−)(−) ON axons (Figure 2). Low power confocal imaging of WT ON cross-section (Figure 2A) shows a central pattern loss of myelin signal (in green), compared with preservation of myelin signal in the center of rNAION-induced SARM1(−)(−) ONs (Figure 2D). Higher power imaging of induced WT ONs (Figure 2B,C) shows that there is a central loss of all myelin (MBP) signals, with axonal degeneration and clumping, shown by neurofilament (SMI312; in red). In the periphery of the ON (Figure 2C), there is a sparing of myelinated axons (Figure 2C: 904 axons), some with a weak myelin signal. The ON shows a progressive axonal loss as one proceeds centrally (the arrow indicates the distance from the ON sheath) and a severe loss of axons centrally (total central axons in photo: 113; all unmyelinated). In contrast, ONs from rNAION-induced SARM1(−)(−) animals show preservation of both individual myelinated axons in the center of the ON lesion (Figure 2E; arrowheads; 423 axons), with sparing of myelinated axons in the periphery of the ON (Figure 2F: 523 myelinated axons; compare with Figure 2C).

### 3.3. Eliminating SARM1 Activity Enhances RGC Somatic Survival after rNAION

We evaluated RGC soma and myelinated axon survival in rNAION-induced, WT-SD, and SARM1(−)(−) animals at 30 d post-induction. Animals used in this portion of the study were initially evaluated for mean optic nerve head (ONH) diameter in the naïve (uninduced) and 2 d post-induced animal, using SD-OCT (Figure 3A). This enabled us to compare the development of relative ONH edema in both WT and SARM1(−) animals. Because we used less than 12 animals for the WT/SARM1 comparison of the edema component, we utilized a Mann–Whitney two-tailed U-test for statistical analysis of OCT values.

All animals included in this portion of the study had a mean optic nerve head (ONH) diameter ≥ 500 μm 2 d post-rNAION induction (threshold ONH edema biomarker for rNAION severity). We previously determined that animals with threshold ONH edema are more likely to have significant (>40%) RGC loss at 30 days post-induction [51]. The contralateral (uninduced) ONHs of the same animals were used as a pre-induction baseline (Figure 3A).

Uninduced (naïve) ONH diameters from SARM1(−)(−) animals were slightly smaller compared with ONHs from WT animals (313.4 ± 10.5 μm (sem); (n = 10 eyes) vs. 336.7 ± 9.63 μm (sem); (n = 7 eyes). This difference was significant (Mann–Whitney two-tailed U-test, *p* = 0.0278). A comparison of ONH diameters following induction also revealed that SARM1(−)(−) animals were slightly smaller, compared with ONHs from WT animals (546.5 ± 14.2 μm (sem); (n = 11 eyes) vs. 566.5 ± 15.6 μm (sem); (n = 8 eyes). However, this difference in 2 d post-rNAION ONH edema was nonsignificant (Mann–Whitney two-tailed test, *p* = 0.136). The rNAION-based relative increase in SARM1(−) ONH diameter is equivalent to that seen in WT ONH. 

We evaluated RGC survival in rNAION-induced WT and SARM1 threshold animals at 30 d post-induction, using Brn3a immunohistochemistry on a flat-mounted whole retinae [51] (Figure 3B). Threshold WT animals showed greater RGC loss compared with SARM1(−) animals (WT: 63.6 ± 3.2% (sem); (n = 15 animals) vs, 49.4 ± 6.8% (sem); (n = 11 animals)). This difference is statistically significant (*p* = 0.049; Mann Whitney one-tailed U-test). We utilized a one-tailed rather than a two-tailed U-test because eliminating SARM1(−) activity has been previously shown to be neuroprotective [35]. Our previously reported study, utilizing similar induction conditions, showed mean values ranging from 71.4 ± 2.4% sem RGC loss when the ≥500 μm threshold effect was considered [51] to 53.2 ± 7.1% sem RGC loss when all animals (subthreshold (<500 μm) as well as threshold ≥ 500 μm) were included. Thus, RGCs from SARM1(−)(−) animals show increased resistance to rNAION-induced acute ischemic optic neuropathy.

### 3.4. Eliminating SARM1 Activity Enhances Long-Term RGC Axonal Survival after rNAION

We also analyzed the appearance of ON tissue late (30 d) after rNAION from WT and SARM1(−)(−) animals. These animals were IVT injected with vehicle (PBS) at the time of induction (Figure 4). Single animals (WT+ SARM1(−)(−)) were used for this part of the study. ON cross-sections were evaluated using antibodies for intact neurofilaments (SMI312) and myelin (MBP). We selected ON specimens that had a similar overall RGC loss and similar Brn3a(+) stereological RGC counts: 286 (WT) and 242 (SARM1(−)(−). As previously noted in the results, Section 3.3, rNAION-induced SARM1(−)(−) animals show a modest overall increase in remaining RGCs compared with WT.

Analysis of ONs from WT animals with severe RGC loss revealed that axonal loss is greatest centrally, with a degree of preservation in the periphery (Figure 4A; outlined in white). SARM1(−)(−) animals also exhibited a similar pattern but had more myelinated axons both peripherally and centrally, as well as apparently scattered through the ischemic central zone, than in WT animals induced to similar levels (Figure 4G, outlined in white). The major difference in SARM1(−)(−) ONs post-vehicle induction appears to be the presence of more intact axons scattered within the primarily ischemic regions and increased myelination scattered within these regions. Thus, eliminating SARM1 activity improves both RGC survival and the quality of the surviving axons after axonal ischemia.

Peripherally, WT animals show a severe loss of axons (SMI312(+) signal, in red) and loss of the myelin (MBP: in green) signal in the affected ON areas of WT rNAION-induced eyes (Figure 4B). A few intact axons remain (arrowheads). This was similar to that seen centrally 5 d post-rNAION induction in WT ONs (compare with Figure 2C), where only a few scattered intact axons remained. In SARM1(−)(−) animals, the myelin signal was preserved to a far greater degree than in WT animals with the same degree of axonal loss (Figure 4D). 

We performed regional axonal quantification for each high-power image (Figure 4B,C,H,I). The SARM1(−)(−) ON exhibits more MBP(+) signals in both peripheral and central regions than the WT ON does. There are fewer intact (SMI312(+)/MBP(+)) myelinated axons in both the WT ON peripheral (high power Figure 4B: 25 myelinated axons) and central (high power Figure 4C: 54 axons) sections, while many more intact, myelinated axons remain in both the periphery (Figure 4H: 82 axons) and center (Figure 4I: 98 axons) of the SARM1(−) ON. Intact representative axons are indicated by arrows. 

rNAION-induced SARM1(−)(−) ONs show an increased number of normally myelinated axons (compare Figure 4F; WT with Figure 4L; SARM1(−)(−), suggesting that reducing or eliminating SARM1 expression in oligodendrocytes also may improve oligodendrocyte preservation. Oligodendrocytes also downregulate MBP expression and undergo prolonged times for apoptosis when axons are lost or damaged [53].

### 3.5. Eliminating SARM1 Activity Synergizes with CNTF-Growth Factor-Mediated, Long-Term Neuroprotection

CNTF administration has been shown to enhance early RGC survival following optic nerve crush and rNAION induction when evaluated early (15 days) post-induction [13]. We wanted to evaluate CNTF’s long-term effect (RGC survival ≥ 30 days) after induction and treatment. We compared CNTF’s effects following rNAION induction in both WT and SARM1(−) animals. Animals were sequentially (1 week) rNAION-induced in both eyes, with the first eye receiving vehicle and the second eye receiving CNTF (see methods). Induction of both eyes enables comparison of each individual’s response to treatment and identification of the degree of RGC loss with each treatment. Each eye received equivalent induction, and RGC stereology was performed 30 days post-second eye induction. Thus, for each animal, the vehicle-treated eye was analyzed at 37 days post-induction, and the CNTF-treated eye was analyzed at 30 days post-induction. Results are shown in Figure 5. Figure 5A shows the relative RGC preservation ratio for the two conditions in WT (white bar: CNTF/Vehicle = 0.97 ± 0.27 sem; n = 11 animals) vs. SARM1(−) (black bar: 1.24 ± 0.20 sem; n = 14 animals). This improvement is statistically significant (*p* < 0.033; Mann–Whitney one-tailed U-test). CNTF by itself does not increase long-term (30 d) RGC survival in rNAION-induced WT eyes, compared with WT eyes injected with vehicle. However, CNTF treatment in rNAION-induced SARM1(−) animals generates a 24% increase in SARM1(−) RGCs, compared with SARM1(−) eyes treated with vehicle. Eliminating SARM1 activity is, by itself, neuroprotective compared with WT (see Figure 3B). Since CNTF by itself does not generate a long-term neuroprotective effect in WT RGCs but combining CNTF with SARM1(−) inactivation enhances RGC survival beyond that of SARM1(−)’s effect alone, this suggests that the combination of CNTF + SARM1(−) provides a synergistic neuroprotective effect.

A representative example of the retinal flat mounts is shown in Figure 5B–E for the two conditions in both WT Figure 5B,C and SARM1(−) animals Figure 5D,E. There were more Brn3a(+) nuclei in the SARM1(−)/CNTF-treated eyes than in equivalently induced WT/CNTF-treated eyes (bar graph comparison, Figure 5A), and there were many more Brn3(+) nuclei (a denser concentration of nuclei) dispersed along the border of maximal RGC loss in the SARM1(−)/CNTF-treated eyes, suggesting that eliminating SARM1 activity enhances RGC survival in axons subjected to relatively ischemic (as opposed to absolute hypoxic) conditions. When the neuroprotective effect of eliminating SARM1 activity alone (14.2% increase in RGCs in SARM1(−) vs. WT; Section 3.4; see also Figure 3B) is added to CNTF’s neuroprotective effect in SARM1(−) animals (24%), the total RGC neuroprotective effect is 38.2%. Thus, combining CNTF with eliminating SARM1(−) activity is synergistically neuroprotective. 

We evaluated axonal loss patterns following CNTF and vehicle-treated nerves in SARM1(−) animals with equivalently induced eyes (Figure 6). A single animal is shown from each group. WT animals treated with IVT vehicle or CNTF showed little difference in the pattern of axonal loss (compare Figure 4A–C, with Figure 6A–C). However, SARM1(−) eyes treated with CNTF showed greatly increased numbers of intact axons in eyes, compared with vehicle (compare Figure 4D–F, with Figure 6D–F). There were many more intact axons scattered through regions with axonal loss in SARM1(−) eyes treated with CNTF and compared with WT eyes either treated with vehicle or with CNTF (compare Figure 6D–F with Figure 4A–C). SARM1(−) ONs treated with CNTF also showed a difference in the axonal loss pattern. Increased regions of preserved, myelinated axons were present in all locations (compare Figure 6E, arrows, with Figure 6B, arrows). SARM1(−) animals show increased numbers of intact SMA312 and MBP signals, even in areas where there were reduced numbers of intact, myelinated axons, compared with WT animals. This difference suggests that loss of SARM1(−) activity increases oligodendrocyte survival, as well as axonal resistance, in regions of partial ischemia. Thus, CNTF treatment of SARM1(−) animals provides an additional synergistic effect that is not seen in WT animals.

## 4. Discussion

### 4.1. Eliminating SARM1 Activity Results in Profound Protection of ON Axons from Ischemia

Ischemic WT axons early post-rNAION induction viewed by TEM show axonal swelling, loss of neurofilaments, and mitochondrial collapse. In contrast, SARM1(−) animals have an essentially normal appearing axoplasm, with preservation of their granularity (compare WT and SARM1(−) centrally located axons in Figure 1). Immunohistologically, this is confirmed by a loss of normal axonal structure, which is shown by the change in pattern of SMI312 immunostaining, with loss of individual axons and axonal swelling in the central ON (Figure 2B). A comparison of myelination patterns reveals more subtle differences. WT axons early in ischemic neuropathy show dissolution and complete loss of normal myelination (Figure 1C; arrows), with complete loss of the MBP signal (compare Figure 2B with Figure 2A, WT). In contrast, SARM1(−) axons show fracturing of the myelin sheaths by TEM, rather than their dissolution and disappearance (compare myelin sheaths in Figure 1C with Figure 1G). The MBP signal is easily detectable in residual SARM1(−) axons in ischemic areas, even where there is extensive axonal loss (compare Figure 2D, normal region, with Figure 2E, ischemic region). Thus, there appears to be a fundamentally different response by SARM1(−) oligodendrocytes to ischemia, compared with WT. SARM1 is present in ON oligodendrocytes, as well as in neurons, and this is demonstrable by sc-seq [54]. However, it is important to note that the simple presence of MBP does not by itself delineate improved oligodendrocyte survival. Future experiments quantifying oligodendrocytes and OPCs using SOX10 will help enable the evaluation of this question.

SARM1 suppression is neuroprotective in axonal ischemia. The parameters utilized for rNAION in the current experiments are predicted to result in a 55–65% RGC loss, in animals induced to threshold (≥500 μm) ONH edema, using the current induction parameters (see methods). A statistically significant reduction in RGC loss was seen in the SARM1(−) animals, compared with WT animals (WT: 63.6% RGC loss vs. SARM1: 49.1% RGC loss; *p* < 0.05, one-tailed test), using the same parameters. SARM1 (−) animals also exhibited a reduced, but nonsignificant degree of ONH edema 2 d post-induction, compared with WT animals induced to the same levels (546.5 μm ± 14.2 SARM1(−) vs. 566.56 ± 15 μm WT). A previous study revealed that in models where there is physical severing of the axon, eliminating SARM1 activity does not render effective protection [40]. Our results suggest that if SARM1 activity is directly responsible for axonal collapse, the elimination of SARM1 can induce significant neuronal protection. 

This suggests that inhibiting or eliminating SARM1 activity is likely to be neuroprotective in conditions involving non-absolute stressors such as partial axonal ischemia. SARM1 suppression is therefore a reasonable approach in early ON neuroprotective treatment.

### 4.2. Combining SARM1 Suppression and CNTF Generates Synergistic Neuroprotection

A surprising outcome is the potent synergistic neuroprotective effect of blocking SARM1 activity combined with CNTF. CNTF treatment of WT animals does not provide long-term protection after rNAION induction (ratio of RGC survival in WT animals between CNTF and vehicle treatment = 0.97 ± 0.27 sem). But CNTF administration coupled with eliminating SARM1 activity significantly increased RGC survival (CNTF/vehicle RGC ratio = 1.24 ± 0.20 sem). There is an increased penumbral density of Brn3a(+) (RGC) nuclei surrounding the region of maximal RGC loss in retinal flat mounts of SARM1(−) animals treated with CNTF, compared with WT animals (compare Figure 5C, WT with Figure 5E, SARM1(−). Future experiments will determine whether this combination both preserves visual function (via acuity and electrophysiologically) and provides even longer-term (60 d or longer) RGC survival.

### 4.3. Mechanisms Responsible for Synergistic Neuroprotection

Unlike axonal transection or crush, NAION-induced axonal ischemia is a non-absolute RGC stress. This may help explain the protective mechanism of SARM1 inhibition, with its ability to enhance RGC survival in conditions that create metastable axons that otherwise would lead to degeneration [55]. CNTF is a potent growth factor that can suppress early RGC death either post-transection [56,57] or in culture [58]. In the previous study, CNTF’s RGC-preservative effect in vivo was measured at 15 days post-induction [23], and this effect is similar to the delayed RGC death effect also reported for brain-derived neurotrophic factor (BDNF) [59,60]. CNTF’s neuroprotective actions may work by stimulation of the ERK/MAPK, Jak-STAT, and PI3K/AKT pathways in neurons [61,62]. Like BDNF, CNTF’s protective effect in rNAION is transitory in wild-type animals. Combining BDNF and CNTF also does not further enhance RGC survival in axotomized animals [63]. But unlike BDNF, CNTF is pro-axogenic [64]. While CNTF does not enhance long-term RGC survival after transection [65], it does enhance RGC survival in a number of less drastic stressors such as elevated IOP [49]. Combining an axogenic growth factor such as CNTF, in SARM1(−) axons that possess resistance to axonal collapse, may increase the expression of axonal repair proteins in these neurons, generating a synergistic mechanism of both resistance to degeneration and stimulation of axonal repair in quiescent axons that would otherwise ultimately lead to neuron degeneration.

Following ON crush or transection, modulating multiple pathways using a combination of treatments that include modifications of the PTEN/mTOR pathway can achieve effective RGC preservation and ON axonal regeneration that had previously been impossible to achieve in vivo [66]. To date, no effective treatments for clinical NAION have been found. However, these previous attempts were all based on single-mechanism approaches. Our current work suggests that, like the recent reports of successful repair in ON crush, the most effective treatment of axonal ischemic conditions may involve proper modulation of multiple mechanisms following axonal ischemia. 

## 5. Conclusions

The current report reveals that SARM1 inhibition or blockade is not only neuroprotective in itself but, by enabling an extended treatment time window prior to axonal collapse, may provide a useful component for enhancing RGC survival and ON recovery following axonal ischemia. SARM1 is a NAD+ cleaving enzyme and inhibitors have been developed that block the SARM1-dependent axon loss [55,67], which may enable further studies of the therapeutic window for SARM1 inhibition, using the rNAION model. The synergistic strategy of combining SARM1 activity inhibition with pro-axogenic growth factors such as CNTF may improve RGC survival and ON recovery in clinical conditions that result in partially compromised axons.

## Figures and Tables

**Figure 1 cells-13-00202-f001:**
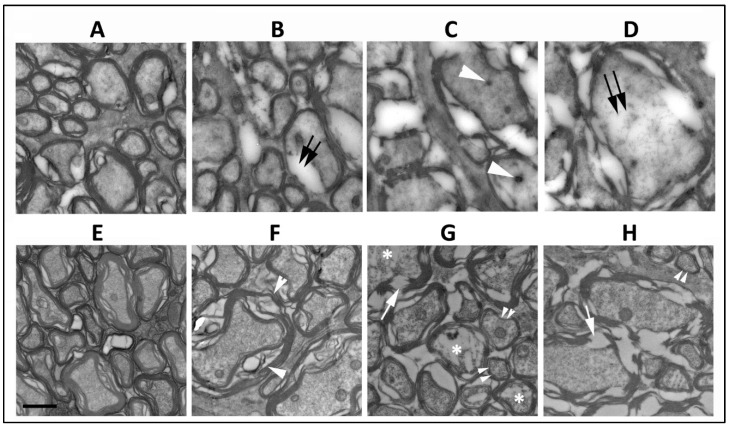
Comparison of RGC axonal structure 5 d post-induction at different distances from the optic nerve periphery. (**A**–**D**) WT ON. (**E**–**H**) SARM1 knockout. (**A**,**E**) Peripheral region. In intact axons and myelination, the axoplasm is granular and myelin is compact. (**B**,**F**) Mid-peripheral region. WT (**B**) intact axons are mingled with axons with degenerating myelin (double arrows) and axoplasm with a loss of granularity and density. Axoplasm in the SARM1(−)(−) ON axons is intact, but some axons have myelin swelling (arrowheads). (**C**,**G**) Central region-1. (**C**) All WT axons are degenerating, and there are dense inclusions in some of the degenerating axons (arrowhead). In contrast, some affected SARM1(−) axons show axoplasmic fragmentation (asterisks), and there is preservation of some small axons (double arrowheads). (**D**,**H**) Central region-2. (**D**) Degenerating WT axons show radiolucent areas (double arrows) and widespread myelin dissolution, while SARM1(−)(−) ONs (**H**) show myelin fragmentation rather than dissolution and relatively intact axoplasm. Scale bar in (**E**) 500 nm.

**Figure 2 cells-13-00202-f002:**
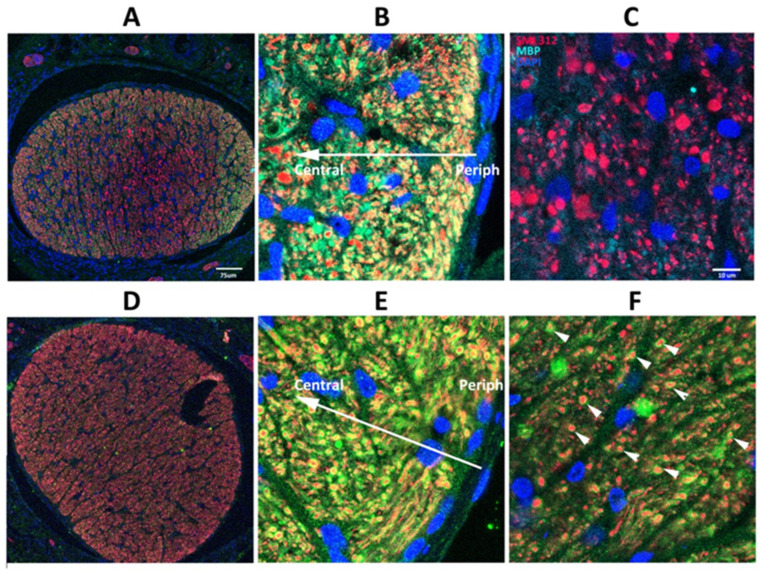
SARM1 loss inhibits early ON axonal and myelin degeneration. 20X magnification: (**A**,**D**) 40X magnification: (**B**,**C**,**E**,**F**). WT: (**A**–**C**) Effects of rNAION on WT ON at 5 d post-induction. (**A**) Low power (20X). There is an uneven distribution of MBP (green) and SMI312/axonal neurofilaments (red), with a central loss of MBP signal, but with preservation of MBP in some of the peripheral areas (arrowheads). (**B**) Peripheral area. Preservation of myelinated axons in the far periphery. There is a progressive loss of intact axons in the more central region (indicated by arrow). (**C**) Central area (40X). Complete MBP loss, with accumulation of large clumps of SMI312 signal and loss of normal axonal distribution. (**D**–**F**) Effects of rNAION on SARM1(−) ON at 5 d post-induction. (**D**) Low power (20X). There is a relatively equal distribution of both MBP and SMI312 signals throughout the nerve. The dark area is a cutting artifact. (**E**) Peripheral region (40X). Preservation of both myelin and axonal neurofilaments, similar to that seen in the WT periphery. The arrow indicates the direction from the periphery to the center. (**F**) Central area (40X). There is a relative loss of SMI312 and MBP signal throughout the region, with scattered individual preserved myelinated axons (arrowheads). The MBP signal is reduced throughout the region but is still present, compared with the WT center (compare (**F**) with (**C**)). Scalebars in (**A**) 75 μm. Scale bar in (**C**) 10 μm.

**Figure 3 cells-13-00202-f003:**
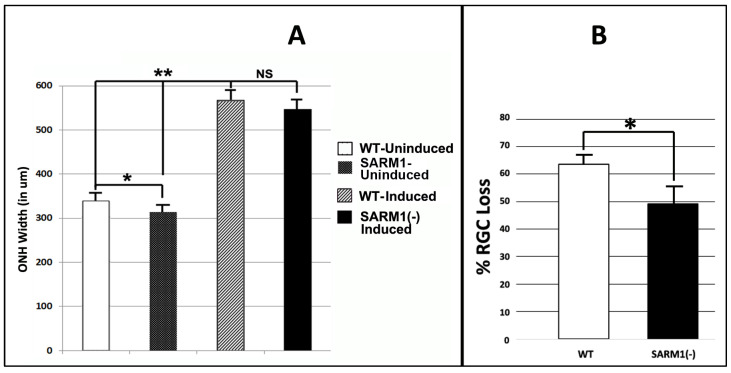
Eliminating SARM1 activity improves RGC survival in ischemic axonopathy. SD-WT animals (black bar) and SARM1 (−)(−) animals (white bar) were rNAION-induced. (**A**) Comparison of mean ONH diameters between uninduced and induced WT and SARM1(−)(−) animals. ONH diameters were determined by SD-OCT. ONH diameters from induced animals were determined 2 d post-rNAION. Naïve ONH diameters were obtained from the contralateral (uninduced) eye of each animal. Naïve ONHs from SARM1(−)(−) animals were reduced in size, compared with WT animals. This was statistically significant (*p* < 0.05; Mann–Whitney two-tailed U-test). Two days post-induction, rNAION results in statistically significant ONH edema-induced expansion, compared with uninduced ONH of either SARM1(−)(−) vs. WT animals (*p* < 0.005, Mann–Whitney two-tailed test). But when the difference between naïve diameters is taken into consideration, the relative difference in ONH edema between rNAION-induced WT and SARM1(−)(−) eyes is nonsignificant (Mann–Whitney two-tailed test, *p* = 0.136). (**B**) Comparison of percent RGC loss in WT vs. SARM1(−)(−) animals. RGC counts were obtained from threshold individuals 30 d post-induction. WT animals show a mean of 63.6 ± 3.2% sem loss, while SARM1(−)(−) animals show a mean of 49.4 ± 6.8% (sem) RGC loss. This difference is statistically significant, *p* < 0.05; Mann–Whitney one-tailed U-test, *p* = 0.049. (* *p* < 0.05; ** *p* < 0.001).

**Figure 4 cells-13-00202-f004:**
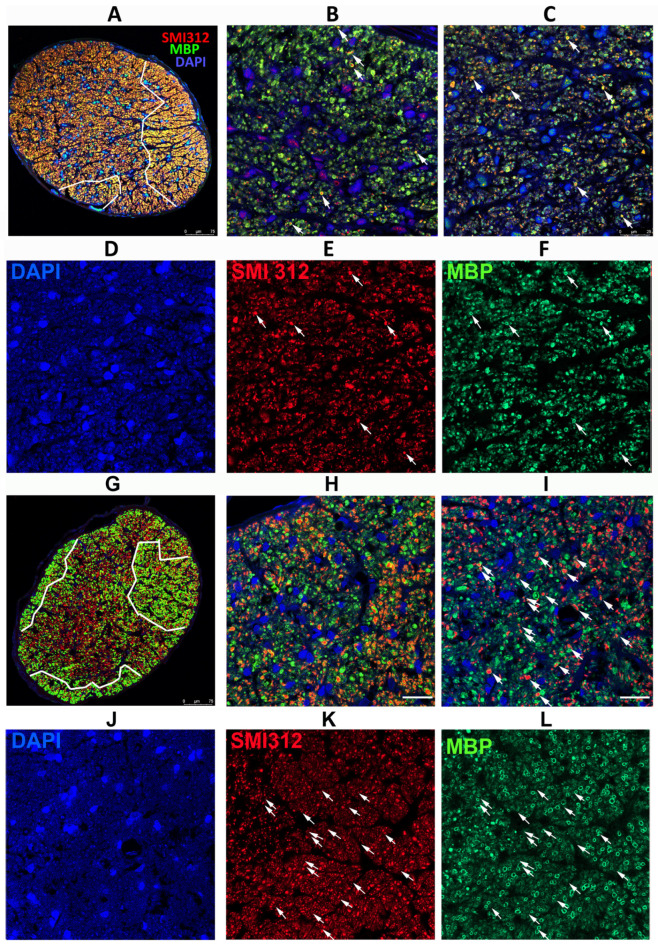
Differences in axonal distribution and myelination in WT and SARM1(−) ONs 30 d post-rNAION induction, following IVT vehicle administration. (**A**–**F**) WT animal. (**G**–**L**) SARM1(−) animal. (**A**,**G**) are 20X (low power) confocal micrographs of mid-distal ON cross-sections. All other sections are 40X (high power) micrographs. (**B**) (WT) and (**H**) (SARM1(−)) are confocal micrographs of the peripheral region. (**C**) (WT) and (**I**) (SARM1(−)) are confocal micrographs of the central ON region. (**D**–**F**) (WT) are the individual channels of (**C**,**J**–**L**) and (SARM1(−)) are the individual channels of (**I**). The arrows in (**C**,**I**) are replicated through the individual channels. All individual channels are shown at the identical confocal settings, to show relative differences of structure. WT ONs exhibit fewer intact axons in the periphery and central regions (arrows), while the SARM1(−)(−) ONs show more intact axons, both as isolated and small patches of intact axons. There are more intact (SMI312-neurofilament) axons from the SARM1(−) ON. SARM1(−) ONs also show better myelin preservation in both peripheral and central regions; this is best seen in the individual (MBP) confocal channel from the central region ((**L**); SARM1(−); compare with (**F**); WT). SMI312 signal is in red, while myelin basic protein (MBP) is in green. DAPI signal in blue. Scale bars: (**A**,**D**) 75 μm. (**H**) 25 μm.

**Figure 5 cells-13-00202-f005:**
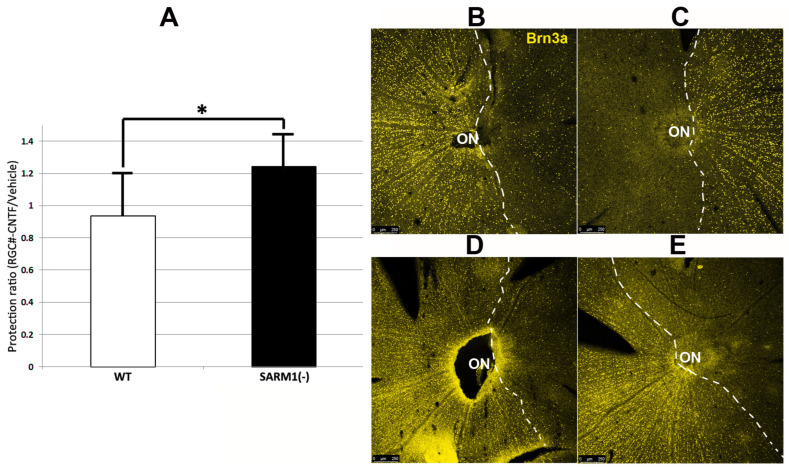
CNTF provides additional RGC-neuroprotective effects in SARM1(−) animals after bilateral rNAION-induction. Both eyes of each animal were rNAION-induced 1 week apart and intravitreally injected sequentially with 2 μL of either vehicle (PBS) or CNTF (25 ng/μL). Surviving RGCs 30 d post-induction from each pair of eyes were quantified and expressed as a protection ratio (CNTF-RGCs/Vehicle-RGCs). (**A**) Mean protection ratios: WT-SD rats (white bar: ratio = 0.94 ± 0.27 sem). SARM1(−) rats (black bar: 1.24 ± 0.20 sem). This difference is statistically significant (*p* = 0.033; Mann–Whitney one-tailed U-test). (**B**–**E**) RGC survival patterns of equivalently induced animals under different conditions. The dashed lines delineate the regions of maximum RGC loss. (**B**,**C**) WT animal. (**B**) Vehicle treated eye. (**C**) CNTF-treated eye. There is reduced cell density in the CNTF-treated eye, compared with the vehicle-treated eye. (**D**,**E**) SARM1(−) animal. (**D**) Vehicle-treated eye. (**E**) CNTF-treated eye. The CNTF-treated eye has an increased number of Brn3a(+) nuclei, compared with the vehicle-treated eye. (*; *p* < 0.05).

**Figure 6 cells-13-00202-f006:**
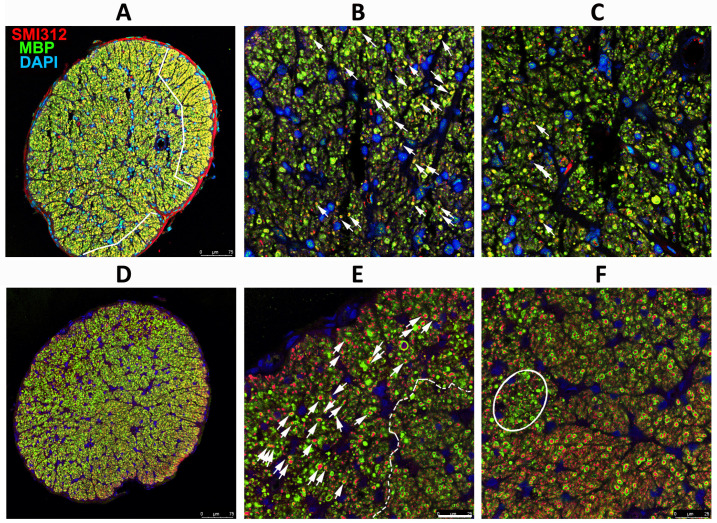
Effects of CNTF after rNAION induction. Axonal (SMI312) and myelin (MBP) patterns. (**A**–**C**) WT+ CNTF ON (RGC stereological count (261). (**D**–**F**) SARM1(−)+ CNTF ON (RGC stereological count 742). (**A**) Low power (20X)/WT. The two areas of maximal intact axons are outlined in white. (**B**) Peripheral affected region. Intact myelinated axons (arrows) are scattered through an area with considerable axonal loss. (**C**) Central region. Few intact axons remain; they are scattered throughout the field. (**D**) Low power/SARM1(−). Many intact axons are present throughout the nerve. While some areas of reduced myelination are present, a strong axonal signal remains. (**E**) SARM1(−) ON peripheral region with an area of axon loss next to a region of intact axons (dashed line). There are still considerable numbers of intact axons (arrows) even in the regions of axonal loss, and the axonal signal is strong. (**F**) SARM1(−) ON central region. Many intact axons are present even in areas of greatest loss, with smaller fields of severe axonal loss (outlined), suggesting increased axonal resistance to ischemia. SMI312: red signal. MBP: green signal. DAPI: blue signal. Scale bars: (**A**,**D**) 75 μm. (**E**) 25 μm.

## Data Availability

All supporting data found in this article is available on request to sbernstein@som.umaryland.edu.

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
