# Peer review of "Synergistic Protection of Retinal Ganglion Cells (RGCs) by SARM1 Inactivation with CNTF in a Rodent Model of Nonarteritic Anterior Ischemic Optic Neuropathy"

_cells, 2024, doi:10.3390/cells13030202_

Round 1

Reviewer 1 Report

Comments and Suggestions for Authors

It is a very well-written paper with a thoughtful experimental design. 

The method section needs to indicate how many rats are used in this experiment in detail.

It will be very exciting to see cross-sectional retinas in both groups stained with retinal ganglion cell markers (with confocal EM) at certain time points after the intravitreal injections of CNTF. 

Author Response

Reviewer 1:

It is a very well-written paper with a thoughtful experimental design. 

  1. The method section needs to indicate how many rats are used in this experiment in detail.

Single animals were used for early 5d WT and SARM1 TEM studies (1+1) and for early (5d) immunohistochemical analyses of ischemic axonal responses. We utilized 7 WT animals and 10 SARM1(-) animals for SD-OCT analysis of ONH edema. A total of 8 WT animals and 11 SARM1(-) animals were used for RGC survival post rNAION (Section 3.3). ON sections from two animals with equivalent amounts of RGC loss from these two groups were used for comparison of axonal loss. A total of 11 WT and 14 SARM1(-) animals were used for the bilateral vehicle/CNTF comparison analyses.  Thus a total of 21 WT animals and 25 SARM1(-) animals were used for all experiments. We state in the methods section:

‘A total of 21 WT animals and 27 SARM1(-) animals were used for all aspects of the study.’

We have also included animal number information for each experiment in the results.

  1. It will be very exciting to see cross-sectional retinas in both groups stained with retinal ganglion cell markers (with confocal EM) at certain time points after the intravitreal injections of CNTF. 

We utilized Brn3a as a marker of ganglion cell (RGC) nuclei, in order to perform quantitative RGC stereology. Brn3a has been shown to be specific for the vast majority of RGCs. We include micrographs RGC immunostaining in the flat mounted retinas (Fig 5B-E).

Reviewer 2 Report

Comments and Suggestions for Authors

The study by Guo and collaborators investigated the protective effects of inhibiting SARM1 activity on retinal ganglion cells (RGCs). Using a model of nonarteritic anterior ischemic optic neuropathy, the authors demonstrated that genetically modified SARM1(-) rats showed suppressed axonal collapse and reduced RGC loss compared to WT rats. Notably, intravitreal injection of CNTF further mitigated RGC loss in SARM1(-) rats. However, there are concerns regarding the ethical standards and research integrity of this study, particularly due to bilateral surgical procedures performed on both eyes. The methodological details are mostly described adequately; however, several experiments lack information about sample sizes, raising uncertainties about the appropriateness of statistical analyses. Additionally, the comparison between CNTF-treated and vehicle-treated eyes is extended by an additional 7 days after rNAION, introducing a potential confounding factor. While the manuscript is intriguing, the assertion that the higher protection observed with combined CNTF treatment is dependent on SARM1(-) raises doubts, as SARM1(-) itself appears to provide a comparable level of protection. The manuscript could benefit from addressing these concerns and providing more clarity to enhance its overall quality.

1. Time gap in rNAION induction (Line 119). (i) The rationale behind inducing rNAION with a one-week interval between eyes is unclear. This raises concerns about the fairness of comparison, as the vehicle-treated eyes with an additional 7 days may exhibit increased RGC loss. (ii) Additionally, performing procedures in both eyes requires clarification regarding ethical approval. An Ethical Statement from ARVO or Research Committee guidelines should be provided.

2. Missing sample sizes. Sample sizes for most experiments are absent, including EM experiments (Fig. 1), Axon loss and demyelination (Fig. 2), and ONH diameters (Fig. 3). Except for Fig. 5 in fig. legend, the “n” is not in material and methods, results or fig. legends.

3. Histogram color and title correction (Fig. 3A). (i) Is the dark gray bar ‘SARM1(-) uninduced’, and the white bar ‘WT-Induced’ instead? And (ii) the color of the dark gray bar is quite similar to the black bar making it difficult to distinguish. Can the authors use a lighter gray color?

4. Quantification of axon loss and demyelination (Fig. 4). While representative examples of axon loss and demyelination are shown in Fig. 4, quantifications or statistics for inter-group comparisons are lacking. It would be beneficial if the authors could delineate, measure, and compare the myelinated-ON area/total-ON area ratio in both groups.

5. Clarity in myelin preservation and RGC axon loss in the ON at 30 days post rNAION induction (line 333, Fig. 4). Images for WT-SD show a higher number of ‘yellow’(red and green signals) axons indicating myelin preservation, while many axons in SARM1(-) are ‘red’ indicating absence of myelin. Then, (i) are all images taken under the same settings? and (ii) I would encourage the authors to include panels with individual channel images for better comparison of axons (red) and myelin (green) in both conditions.

6. DAPI signal and glial populations (Fig. 4). Have the authors noticed that the SARM1(-) ON at 30 days post rNAION induction seems to have a lower number of DAPI nuclei (compare Fig 4C vs 4F)? Have the authors evaluated the glial populations in the ON (oligodendrocytes, astrocytes, and microglia cells)?

7. Figure legends often repeat detailed information present in corresponding result sections. E.g., Figure 3 legend (257-271) and main text (272-278). To enhance clarity, the legends should focus on providing additional details and avoid duplicating information found in the results.

8. ONH diameter comparison (Line 493). The authors discuss that the ONH diameter after rNAION induction in SARM1(-) were not significant but slightly smaller. I would suggest comparing if the ONH diameter in SARM1(-) is proportionally smaller than before inducing rNAION.

9. Evaluation of SARM1 inhibitors. Considering the potential drawbacks of inducing SARM1 knockout in patients, have the authors explored the use of any SARM1 inhibitors as an alternative therapeutic approach?

10. Contradictory CNTF-induced RGC survival statements. If CNTF has proven to be neuroprotective for RGCs after rNAION (line 95), why the ratio of RGC survival in the CNTF-treated WT retinas is similar to vehicle-treated (line 413, Fig. 5A)? How can the authors explain it? Note that it has been previously reported that CNTF can protect during short, but no larger periods of time (e.g., PMID: 19268467).

11. RGC survival rates significance (Figs 3B and 5A). Can the authors compare the % of surviving RGCs after the synergistic neuroprotective effect of SARM1(-) and CNTF vs. the % of surviving RGCs in SARM1(-) (with respect to their corresponding WT groups, Figs. 5A and 3B)?

12. Discussion on Synergistic Neuroprotection. Could the authors expand more on the mechanisms underlying this synergistic neuroprotective effect of SARM1(-) and CNTF? E.g. E.g., a previous report suggested that cyclic AMP potentiates ciliary neurotrophic factor (PMID: 12595238).

13. Vision improvement. Have the authors conducted any functional or behavioral tests to assess improvements in vision resulting from the combined treatment?

Minor Comments.

14. Brn3a was detected by the corresponding Cy3 secondary antibody (line 136). Although using a confocal the images are in grayscale, the images for Brn3a in Fig. 5 (page 10) are green (which corresponds to 488 fluorophore).

15. Please add the corresponding references to the sentences in

-line 196. “crush or section (ref), but suppressing”

-lines 360-361. “(Mathews reference)”

-lines 496-497. “A previous study revealed that in models where there is the physical severing of the axon, eliminating SARM1 activity does not render effective protection”

16. I would recommend adding the term “Fig.” to the Figure calls in the main text. E.g., line 200: (2A), line 203: (2B-C), line 205: (2C), etc

17. Please revise the text for different typos:

-Line 293. “(MBP).e selected

-Line 295. “noted in results section 3, rNAION-induced”; do de authors mean section 3.3?

-Line 516. “increased penumbra of Brn3a(+)...”; do the authors mean ‘increased number of Brn3a(+)?

18. English need proofreading, e.g., expressions like:

-Line 361. “times distant (>21...”; Do the authors mean later time points, later stages...?

19. Relevant reference:

-Finnegan LK, Chadderton N, Kenna PF, Palfi A, Carty M, Bowie AG, Millington-Ward S, Farrar GJ. SARM1 Ablation Is Protective and Preserves Spatial Vision in an In Vivo Mouse Model of Retinal Ganglion Cell Degeneration. Int J Mol Sci. 2022 Jan 30;23(3):1606. doi: 10.3390/ijms23031606. PMID: 35163535; PMCID: PMC8835928.

Comments on the Quality of English Language

Minor errors and typos

Author Response

Reviewer #2

Comments and Suggestions for Authors

The study by Guo and collaborators investigated the protective effects of inhibiting SARM1 activity on retinal ganglion cells (RGCs). Using a model of nonarteritic anterior ischemic optic neuropathy, the authors demonstrated that genetically modified SARM1(-) rats showed suppressed axonal collapse and reduced RGC loss compared to WT rats. Notably, intravitreal injection of CNTF further mitigated RGC loss in SARM1(-) rats. However,

  1. There are concerns regarding the ethical standards and research integrity of this study, particularly due to bilateral surgical procedures performed on both eyes.

This was reviewed by the UMB IACUC, and was passed because the rNAION procedure does not blind the animal’s eye, but simply reduces overall function.  We have included the statement in the methods section, rNAION induction:

‘For vehicle vs CNTF treatment, animals of both strains were sequentially induced in both eyes, one week apart. Because the rNAION procedure reduces visual function, but does not blind the animal, the IACUC committee found that it was ethically appropriate to treat both eyes of the same animal in the bilateral vehicle/CNTF treatment experiment, and reduced the overall number of animals required for analysis.’

  1. The methodological details are mostly described adequately; however, several experiments lack information about sample sizes, raising uncertainties about the appropriateness of statistical analyses.

We have included (section 3.3): We utilized 11 SARM1(-) and 15 WT animals for this part of the study.

We also include in 3.3:

‘Because we used less than 12 animals for the SARM1 component, we utilized a Mann-Whitney two tailed U test for statistical analysis of OCT values.’

And for analysis of the effect of  eliminating SARM1 activity in RGC neuroprotection:

‘We utilized a one-tailed, rather than two tailed U test because eliminating SARM1(-) activity has been previously shown to be neuroprotective.’

  1. The comparison between CNTF-treated and vehicle-treated eyes is extended by an additional 7 days after rNAION, introducing a potential confounding factor.

We apologize for the confusion. The bilateral treatment approach is an important design point. The bilateral treatment approach is the optimal way to determine comparative effectiveness, since it reduces the inter-individual response difference between two treatments, and can help uncover the actual protective effect of drugs. Comparative effectiveness analysis is used in human drug trials (treatment A>>washout>> treatment B). We minimized the potential effect of induction and intravitreal injection stress on the second eye by using a 1 week (7 half-lives vitreous turnover) washout interval between treatments. By administering CNTF after the second eye induction and utilizing intravitreal treatment, we minimized potential neuroprotective effects of CNTF on the contralateral (vehicle) eye. This is shown by the mean neuroprotective effect only on the CNTF treated eye. While different washout periods can be used, we chose a washout period that exceeded the initial stress response time (Bernstein et al, 2003), yet enabled euthanasia for collection of long-term post-induction tissue from both eyes.  We have completely rewritten the ‘Post-induction intravitreal injection’ section in methods,

‘Design of CNTF comparative treatment effectiveness trial for WT and SARM1(-) animals: We utilized a bilateral treatment approach to determine comparative neuroprotective effectiveness of vehicle (PBS) and CNTF, since it reduces the inter-individual response difference between two treatments and can help uncover the actual protective effect of drugs. We minimized the potential effect of induction and intravitreal injection stress on the second eye by using a 1 week washout interval between treatments, and by using placebo for treatment of the first induced eye. Following rNAION induction of the first eye , animals were intravitreally injected unilaterally with 2ul of vehicle (PBS, pH 7.4). The second eye of the same animal was induced one week after induction of the first eye and injected with 2ul of 50ng CNTF in PBS. Both WT (SD) and SARM1 animals were compared.’

And the figure legend 5 title has been changed to:

Figure 5. CNTF provides additional RGC-neuroprotective effects in SARM1(-) animals after bilateral rNAION-induction

  1. While the manuscript is intriguing, the assertion that the higher protection observed with combined CNTF treatment is dependent on SARM1(-) raises doubts, as SARM1(-) itself appears to provide a comparable level of protection.

This is an important point. Eliminating SARM1 activity by itself is neuroprotective (Figure 3B). CNTF provides ADDITIONAL neuroprotection, but only in animals without SARM1 activity. The total neuroprotective effect of eliminating SARM1 activity (14%; difference in 3B) and adding CNTF (ratio difference in Figure 5A=24%)  is 38%. Thus, the neuroprotective effect of eliminating SARM1 activity and adding CNTF is synergistic. It is synergistic rather than additive, since CNTF by itself in wild type provides no protection at all. This is stated in section 3.5:

‘When the neuroprotective effect of eliminating SARM1 activity alone (14.2% increase in RGCs in SARM1(-) vs WT; section 3.4; also see Figure 3B), is added to CNTF’s neuroprotective effect in SARM1(-) animals (24%), the total RGC neuroprotective effect is 38.2%. Thus, combining CNTF with eliminating SARM1(-) activity is synergistically neuroprotective.’

  1. The manuscript could benefit from addressing these concerns and providing more clarity to enhance its overall quality.
  2. Time gap in rNAION induction (Line 119). (i) The rationale behind inducing rNAION with a one-week interval between eyes is unclear. This raises concerns about the fairness of comparison, as the vehicle-treated eyes with an additional 7 days may exhibit increased RGC loss. (ii) Additionally, performing procedures in both eyes requires clarification regarding ethical approval. An Ethical Statement from ARVO or Research Committee guidelines should be provided.

We have addressed these concerns:

(i) In Methods, we include: Design of CNTF comparative treatment effectiveness trial for WT and SARM1(-) animals: We utilized a bilateral treatment approach to determine comparative neuroprotective effectiveness of vehicle (PBS) and CNTF, since it reduces the inter-individual response difference between two treatments, and can help uncover the actual protective effect of drugs. We minimized the potential effect of induction and intravitreal injection stress on the second eye by using a 1 week washout interval between treatments, and by using placebo for treatment of the first induced eye.

(ii) In Methods, we include in rNAION induction: Because the rNAION procedure reduces visual function, but does not blind the animal, the IACUC committee found that it was ethically appropriate to treat both eyes of the same animal in the bilateral vehicle/CNTF treatment experiment, and reduced the overall number of animals required for analysis.

  1. Missing sample sizes. Sample sizes for most experiments are absent, including EM experiments (Fig. 1).

We include in Methods the modified sectionTransmission electron microscopy: Single animals (WT and SARM1(-) were used for this part of the study, since only the axonal structure changes at specific times were evaluated, and axon numbers were not quantified.

  1. Axon loss and demyelination (Fig. 2).

We have included a new section in Methods,  ‘Early axonal loss and demyelination analysis: Single animals (both WT and SARM1(-) were used for this part of the analysis, since only a gross comparison of relative axonal preservation and myelination differences between WT and SARM1(-) nerves at early (5d) times during ischemic conditions was desired. ON sections were reacted with SMI312 and anti-MBP antibodies (see immunohistology) and relative number of intact, myelinated axons were quantified in the individual sections.’

  1. and ONH diameters (Fig. 3). Except for Fig. 5 in fig. legend, the “n” is not in material and methods, results or fig. legends.

We have included in the text (section 3.3):

‘Uninduced (naïve) ONH diameters from SARM1(-) animals were slightly smaller compared with ONHs from WT animals (313.4±10.5um (sem); (n=10 eyes) vs (336.7±9.63um; (sem)(n=7 eyes)). This difference was significant (two tailed Mann-Whitney U-test, p=0.0278). A comparison of ONH diameters following induction also revealed that SARM1(-) animals were slightly smaller, compared ONHs from WT animals (546.5±14.2um (sem);(n=11 eyes) vs 566.5±15.6um (sem); (n=8 eyes). However, this difference in 2d post-rNAION ONH edema was nonsignificant (Mann-Whitney two-tailed test, p=0.136).’

We also add:

We evaluated RGC survival in rNAION-induced WT- and SARM1 threshold animals at 30d post-induction, using Brn3a immunohistochemistry on flat mounted whole retinae ([31] (Fig 3B). Threshold WT animals showed greater RGC loss compared with SARM1(-) animals (WT: 63.6±3.2% (sem); (n=8 animals) vs, 49.4±6.8% (sem); (n=11 animals). This difference is statistically significant (p=0.049; Mann Whitney 1-tailed U test). We utilized a one-tailed, rather than two tailed U test because eliminating SARM1(-) activity has been previously shown to be neuroprotective Reference: Liu, P. et al. Differential effects of SARM1 inhibition in traumatic glaucoma and EAE optic neuropathies. Molecular Therapy-Nucleic Acids 32, 13-27, doi:10.1016/j.omtn.2023.02.029 (2023).

We have also included this data in the figure 3 legend.

  1. Histogram color and title correction (Fig. 3A). (i) Is the dark gray bar ‘SARM1(-) uninduced’, and the white bar ‘WT-Induced’ instead? And (ii) the color of the dark gray bar is quite similar to the black bar making it difficult to distinguish. Can the authors use a lighter gray color?

We have changed the histogram color to a lighter gray color in Figure 3A to make them easier to distinguish.

  1. Quantification of axon loss and demyelination (Fig. 4). While representative examples of axon loss and demyelination are shown in Fig. 4, quantifications or statistics for inter-group comparisons are lacking. It would be beneficial if the authors could delineate, measure, and compare the myelinated-ON area/total-ON area ratio in both groups.

We have tried to quantify the differences in myelination and axonal distribution, and have found this quite difficult, utilizing both densitometric methods and area counts. There are subtle shades in the ON from degenerating axons, and MBP(+) signals that are not associated with intact (SMI(+)/SMI312(+) axons. In addition, the subtle shade differences between intact and degenerating axons requires direct observer discrimination  at high power, rather than densitometry. Please see our reply below in point 5.

To solve this problem, we manually counted the axons in the high power fields from WT and SARM1(-) ONs. We report this in the Figure 4 legend:

Figure 4. Differences in axonal distribution and myelination in WT and SARM1(-) ONs 30d post-rNAION induction, following IVT vehicle administration. A-C: WT animal. D-F: SARM1(-) animal. A, D: Low power (20X) confocal micrographs of mid-distal ON cross-sections. Intact regional quantification results from an ON from each group. There is regional preservation of axons in both strains, but the SARM1(-) ON exhibits more MBP(+) signal in both peripheral and central regions. B,E: Peripheral affected region (40X). WT ONs exhibit few intact axons in their depleted regions, while the SARM1(-) ONs show both individual and patches of intact, myelinated axons (E: in brackets). C,F: Central region (40X). There are fewer intact (SMI312(+)/MBP(+) myelinated axons in the WT ON peripheral (high power Fig 4B: 25 myelinated axons) and central (high power Fig 4C: 54 axons) sections, while many more (128%) intact, myelinated axons remain in both the periphery (high power Fig 4E: 82 axons) and center (High power Fig 4F: 98 axons) of the SARM1(-) ON (compare axons indicated by arrows in C and F). SMI312 (intact neurofilaments) signal is in red, while myelin basic protein (MBP) is in green. DAPI signal in blue. Scale bars: A,D: 75um. E: 25um.

In the results section:

Analysis of ONs from WT animals with severe RGC loss revealed that axonal loss is greatest centrally, with a degree of preservation in the periphery (Fig 4A; outlined in white). SARM1(-) animals also exhibited a similar pattern, but more myelinated axons were found extending both peripherally and centrally, as well as apparently scattered through the ischemic central zone, than in WT animals induced to similar levels (4D, outlined in white).

  1. Clarity in myelin preservation and RGC axon loss in the ON at 30 days post rNAION induction (line 333, Fig. 4). Images for WT-SD show a higher number of ‘yellow’(red and green signals) axons indicating myelin preservation, while many axons in SARM1(-) are ‘red’ indicating absence of myelin. Then, (i) are all images taken under the same settings? and (ii) I would encourage the authors to include panels with individual channel images for better comparison of axons (red) and myelin (green) in both conditions.

  1. We have done so. We have made a new figure showing the individual confocal frames for DAPI, SMI312/axons and MBP/Myelin, and all are at the same settings. It is apparent in these frames that there are increased intact myelinated axons in the periphery (E and F) and that there are many fewer intact/complete myelin sheaths in the WT ON after rNAION at 30 days than in the SARM1(-) ON (compare frames B and C: WT, with E and F: SARM1(-)). This new figure is particularly informative when comparing the MBP frames.

  1. DAPI signal and glial populations (Fig. 4). Have the authors noticed that the SARM1(-) ON at 30 days post rNAION induction seems to have a lower number of DAPI nuclei (compare Fig 4C vs 4F)? Have the authors evaluated the glial populations in the ON (oligodendrocytes, astrocytes, and microglia cells)?

While this is an interesting and important thought, we have not included individual analyses of all glial elements (GFAP(+) astrocytes, IBA1(+) inflammatory cells, CD68/extrinsic macrophages and CCR2 microglia, Olig2/SOX10 OIPCs and oligodendrocytes), because this is a entire separate study that will require an intensive future article with analysis both by qPCR and immunohistology.

  1. Figure legends often repeat detailed information present in corresponding result sections. E.g., Figure 3 legend (257-271) and main text (272-278). To enhance clarity, the legends should focus on providing additional details and avoid duplicating information found in the results.

We have eliminated the repeated data from the figure legends (Fig 3), simplified the legend, and placed the numerical data in the results section 3.3 paragraph below the Figure legend.

  1. ONH diameter comparison (Line 493). The authors discuss that the ONH diameter after rNAION induction in SARM1(-) were not significant but slightly smaller. I would suggest comparing if the ONH diameter in SARM1(-) is proportionally smaller than before inducing rNAION.

 We include: ‘The rNAION-based relative increase in SARM1(-) ONH diameter is equivalent to that seen in WT ONH.’

  1. Evaluation of SARM1 inhibitors. Considering the potential drawbacks of inducing SARM1 knockout in patients, have the authors explored the use of any SARM1 inhibitors as an alternative therapeutic approach?
  • This is our ultimate objective, however the vast majority of published SARM1 inhibitors have only been tested in vitro. The few SARM1 inhibitors tested in vivo were used in short-term studies and were much less effective than the genetic knockout (e.g. Bosanac, PMID 33964142, Bratkowski, PMID 36087583). Moreover, these inhibitors are not commercially available. As such, we chose the genetic KO as the most appropriate reagent for testing the role of SARM1 in NAION at this time.     
  1. Contradictory CNTF-induced RGC survival statements. If CNTF has proven to be neuroprotective for RGCs after rNAION (line 95), why the ratio of RGC survival in the CNTF-treated WT retinas is similar to vehicle-treated (line 413, Fig. 5A)? How can the authors explain it? Note that it has been previously reported that CNTF can protect during short, but no larger periods of time (e.g., PMID: 19268467).

We apologize for the confusing statement. This is an important point. CNTF by itself does not increase long-term (30d) RGC survival in WT animals. CNTF provides additional long-term RGC neuroprotection in SARM1(-) animals, complementing SARM1(-) potentiation of  survival in damaged, but metastable axons that would otherwise degenerate{Hughes, 2021 #8310}. Thus, CNTF treatment further enhances RGC survival only in cells which (in this case) have the added neuroprotective SARM1(-) advantage. This is the essence of a synergistic effect: the addition of a second agent enhances survival far beyond that of the single agent alone. We have revised this paragraph for clarity and explanation:

‘Results are shown in Figure 5. 5A shows the relative RGC preservation ratio for the two conditions in WT (white bar: CNTF/Vehicle=0.97±0.27sem) vs SARM1(-) (black bar: 1.24±0.20 sem). CNTF by itself does not increase long-term (30d) RGC survival in rNAION-induced WT eyes, compared with WT eyes injected with vehicle. However, CNTF-treatment in rNAION-induced SARM1(-) animals generates a 24% increase in SARM1(-) RGCs, compared with SARM1(-) eyes injected with vehicle. Eliminating SARM1 activity is itself neuroprotective, compared with WT (see Fig.3B). Since CNTF by itself does not generate a long-term neuroprotective effect in WT RGCs, but combining CNTF with SARM1(-) inactivation enhances RGC survival beyond that of SARM1(-)’s effect alone, this suggests that the combination of CNTF + SARM1(-) provides a synergistic neuroprotective effect.’

  1. RGC survival rates significance (Figs 3B and 5A). Can the authors compare the % of surviving RGCs after the synergistic neuroprotective effect of SARM1(-) and CNTF vs. the % of surviving RGCs in SARM1(-) (with respect to their corresponding WT groups, Figs. 5A and 3B)?

We have done so, and delineate CNTF’s synergistic effect in SARM1(-) animals, compared with the WT animals (Page 11, and first paragraph of page 12). We rewrote the Fig 5 legend for clarity.

  1. Discussion on Synergistic Neuroprotection. Could the authors expand more on the mechanisms underlying this synergistic neuroprotective effect of SARM1(-) and CNTF? E.g. E.g., a previous report suggested that cyclic AMP potentiates ciliary neurotrophic factor (PMID: 12595238).

 We thank the reviewer greatly for pointing this weakness out. We have extensively revised and reduced this section to include a more focused discussion of synergistic neuroprotection. The revised section reads:

4.3 Mechanisms responsible for synergistic neuroprotection. Unlike axonal transection or crush, NAION-induced axonal ischemia is a non-absolute RGC stress. This may help explain the protective mechanism of SARM1 inhibition, with its ability to enhance RGC survival in conditions that create metastable axons that otherwise would lead to degeneration{Hughes, 2021 #8310}. CNTF is a potent growth factor that can suppress early RGC death either post-transection [35] [36], or in culture[37]. In the past, this effect in vivo was typically measured at 15 days post-insult[13], and is similar to the delayed RGC death effect also reported for brain derived neurotrophic factor (BDNF) {Sendtner, 1994 #9148}. CNTF may work by stimulation of the ERK/MAPK, Jak-STAT and PI3K/AKT pathways in neurons[29,44]. Like BDNF, the CNTF effect is transitory in wild type animals and combining these neurotrophins also does not further enhance survival{Mey, 1993 #8612}. However, CNTF is also pro-axogenic, and while it does not enhance long-term RGC survival after transection [38], it does enhance RGC survival in a number of less drastic stressors such as elevated IOP [39]. The addition of an axogenic growth factor such as CNTF, in axons with resistance to axonal collapse, may increase expression of axonal repair proteins in these neurons, generating a synergistic mechanism of both resistance to degeneration and stimulation of axonal repair in quiescent axons that would otherwise ultimately lead to neuron degeneration.

Following ON crush or transection, modulating multiple pathways using a combination of treatments that include modifications of the PTEN/mTor pathway can achieve effective RGC preservation and ON axonal regeneration that had previously been impossible to achieve in vivo[40]. To date, no effective treatments for clinical NAION have been found. However, these previous attempts were all based on single mechanism approaches. Our current work suggests that, like the recent reports of successful repair in ON crush, the most effective treatment of axonal ischemic conditions may involve proper modulation of multiple mechanisms following axonal ischemia.

We have also revised the Conclusions section to follow the revised ms more closely.

  1. Vision improvement. Have the authors conducted any functional or behavioral tests to assess improvements in vision resulting from the combined treatment?

  No, we did not. The reason for this is that the SARM1(-) animals are on Sprague-Dawley (SD) albino background, which have reduced visual acuity and abnormal VEP responses compared with normally pigmented (eg Long-Evans or Brown Norway) animals. The need for breeding the SARM1(-) mutation into a pigmented strain would have required at least 5 generations of back-crossing to obtaining a stable genetic background in pigmented animals. We are planning a new study with to answer this question. We include the statement in the methods section under animals:

‘Because the SARM1(-) strain is on an SD-albino background, and albino animals have reduced visual acuity compared with pigmented animals, we utilized direct anatomical analysis using RGC quantification and ON histology, rather than functional tests.’

Minor Comments.

  1. Brn3a was detected by the corresponding Cy3 secondary antibody (line 136). Although using a confocal the images are in grayscale, the images for Brn3a in Fig. 5 (page 10) are green (which corresponds to 488 fluorophore).

We have corrected the color in the figure to Cy3 emission (yellow).

  1. Please add the corresponding references to the sentences in

-line 196. “crush or section (ref), but suppressing”

included (new reference [22]’

-lines 360-361. “(Mathews reference)”

‘Included’

-lines 496-497. “A previous study revealed that in models where there is the physical severing of the axon, eliminating SARM1 activity does not render effective protection”

included. ‘[22]’ 

  1. I would recommend adding the term “Fig.” to the Figure calls in the main text. E.g., line 200: (2A), line 203: (2B-C), line 205: (2C), etc

  1. Please revise the text for different typos:

-Line 293. “(MBP).e selected”

revised

-Line 295. “noted in results section 3, rNAION-induced”; do de authors mean section 3.3?

revised. Thank you.

-Line 516. “increased penumbra of Brn3a(+)...”; do the authors mean ‘increased number of Brn3a(+)?

 revised to: ‘an increased penumbral density of Brn3a(+) (RGC) nuclei’

  1. English need proofreading, e.g., expressions like:

-Line 361. “times distant (>21...”; Do the authors mean later time points, later stages...?

 revised to: ‘survival has not been measured at later time points than 21 days from lesion induction.’

  1. Relevant reference:

 -Finnegan LK, Chadderton N, Kenna PF, Palfi A, Carty M, Bowie AG, Millington-Ward S, Farrar GJ. SARM1 Ablation Is Protective and Preserves Spatial Vision in an In Vivo Mouse Model of Retinal Ganglion Cell Degeneration. Int J Mol Sci. 2022 Jan 30;23(3):1606. doi: 10.3390/ijms23031606. PMID: 35163535; PMCID: PMC8835928.

Thank you! This is an excellent reference! We have included this information in methods/animals, because we are currently using albino animals for the current study and it does not affect the current results. We state:

‘It should be noted that deleting SARM1 activity has also been shown to preserve spatial vision in toxic optic neuropathies in pigmented animals[30] ‘.

We have also eliminated two references by the Bernstein lab group (Bernstein 2022 and Goldenberg-cohen 2002), to reduce the self-referencing. These were not absolutely needed. The remaining references are critical for the integrity of the manuscript.

Additional Changes:

  1. changed abstract to clarify CNTF’s additive effect
  2. Introduction: However, these individual factors neuroprotective effects are modest even when administered close to lesion onset, and CNTF’s RGC-neuroprotective effects were only quantified at 15d post-induction [13].

Round 2

Reviewer 2 Report

Comments and Suggestions for Authors

In this updated version, the authors have conscientiously addressed the majority of concerns raised in the previous review. Responding to the reviewer's suggestions, the authors have introduced new images and meticulously re-edited the main text, incorporating clarifications in figure legends and additional comments in the discussion. Consequently, the manuscript has undergone significant improvement. However, a few issues require attention, as outlined below:

1. Histogram (Fig. 3A): Despite the authors rectifying the colors as per the previous recommendation (#3), there remains an oversight in the naming of bars in Fig. 3A. The labels for 'white and light gray bars' erroneously still refer to 'WT-Uninduced,' while 'dark gray and black' are incorrectly labeled as 'SAPRM1(-) Induced.' It is suggested that 'Dark gray' be revised to “SARM1(-) Uninduced,” and 'light gray' to “WT-induced.”

2. Fig. 4 Images (Lines 412-413): The inclusion of new images (K and L) supporting the conclusions in Fig. 4 is appreciated. However, a discrepancy is noted concerning the assertion that images K and L (SMI312 and MBP, respectively) correspond to J and I (merged and DAPI, respectively). A careful reevaluation of these images is recommended.

3. Time Points Post-rNAION (Lines 448-453): Despite the authors' efforts to elucidate the "Post-induction intravitreal injection method," I am still confused regarding the time points post-rNAION for the Vehicle and CNTF groups. It is essential to clarify whether both groups were analyzed at 30 days post-rNAION or if the Vehicle group was assessed at 37 days and the CNTF group at 30 days post-rNAION. This distinction is crucial for ensuring the comparability of the groups, considering the potential impact of longer time points on RGC degeneration.

4. Oligodendrocyte Preservation (Lines 435-436): The claim of higher oligodendrocyte preservation lacks supporting data in the presented experiments, as myelin preservation is not exclusively dependent on oligodendrocyte viability. To support this assertion, specific staining for oligodendrocytes would be recommended (Oligo2...).

Minor Edits and Typos:

Lines 341-345: Considering that the reference pertains to the authors, it is advisable to state: "In our previous studies..."

Lines 165-167: Please review this sentence; an open parenthesis is present without a corresponding closed parenthesis.

Lines 338-339: Please review this sentence; an open parenthesis is present without a corresponding closed parenthesis.

Line 223: Add the term ‘Fig’ to “(1G...”

References that can be considered.    

Lines 174-175: Regarding Brn3a as a reliable marker for RGCs, a reference could help: 19264888, 36594396.

Line 640: a study focused on axonal damage in RGCs has demonstrated the protective effect of BDNF in rats: 21315070.

Comments on the Quality of English Language

The English sounds

Author Response

We thank reviewer 2 for his meticulous (and extremely helpful!) review of our manuscript. We have made the appropriate requested changes and hope that this version improves the quality by eliminating errors and satisfies their questions. The responses are in italics. We have also added two additional references as requested by the reviewer (Nadal-Nicolas et al. and Migallon-Sanchez et al.), to substantiate both the use of Brn3a and the effect of BDNF. Please see below.

We have attached the final (all changes accepted) version. And Happy Holidays.

Very truly yours,

Steve Bernstein

For the Authors

In this updated version, the authors have conscientiously addressed the majority of concerns raised in the previous review. Responding to the reviewer's suggestions, the authors have introduced new images and meticulously re-edited the main text, incorporating clarifications in figure legends and additional comments in the discussion. Consequently, the manuscript has undergone significant improvement. However, a few issues require attention, as outlined below:

  1. Histogram (Fig. 3A): Despite the authors rectifying the colors as per the previous recommendation (#3), there remains an oversight in the naming of bars in Fig. 3A. The labels for 'white and light gray bars' erroneously still refer to 'WT-Uninduced,' while 'dark gray and black' are incorrectly labeled as 'SAPRM1(-) Induced.' It is suggested that 'Dark gray' be revised to “SARM1(-) Uninduced,” and 'light gray' to “WT-induced.”

Thank you for catching this error! This has been revised.

  1. 4 Images (Lines 412-413): The inclusion of new images (K and L) supporting the conclusions in Fig. 4 is appreciated. However, a discrepancy is noted concerning the assertion that images K and L (SMI312 and MBP, respectively) correspond to J and I (merged and DAPI, respectively). A careful reevaluation of these images is recommended.

We have re-evaluated the images. The problem was that SMI312/red signal and MBP/green signal was switched (red was green and vice versa) in the 63X (Figure 4 B-C and Figure 4 H-I) panels. We have changed the panels so that they indeed represent the appropriate color signals throughout (C is appropriately represented in the individual panels D-F and I is appropriately represented in J-L).

  1. Time Points Post-rNAION (Lines 448-453): Despite the authors' efforts to elucidate the "Post-induction intravitreal injection method," I am still confused regarding the time points post-rNAION for the Vehicle and CNTF groups. It is essential to clarify whether both groups were analyzed at 30 days post-rNAION or if the Vehicle group was assessed at 37 days and the CNTF group at 30 days post-rNAION. This distinction is crucial for ensuring the comparability of the groups, considering the potential impact of longer time points on RGC degeneration.

While this is a useful caveat and important for descriptive clarity, we previously have  found that, in WT animals, the vast majority of RGC loss occurs by 21 days post-induction (see the Slater et al, IOVS 2008 reference). The presence of CNTF (or BDNF) can extend RGC survival in WT animals, as measured by Brn3a {Cen et al, 2007 IOVS 48}{Sanchez-Migallon et al, 2016 IOVS 57}, but only for 2-3 days. The fact that there is no improvement in RGC survival in WT animals treated with CNTF (post-30d), as compared with vehicle (post-37d), suggests that it is the coupling of the loss of SARM1 activity, with CNTF, that provides the enhanced effect. In studies that utilize both eyes, a new study incorporating times even 60d post-induction would still have some relative difference between the two eyes.

We have added in section 3.5: ‘We wanted to evaluate CNTF’s long-term effect (RGC survival ≥30 days after induction and treatment.’

We further add: ‘Thus, for each animal the vehicle-treated eye was analyzed at 37 days post-induction, and the CNTF-treated eye was analyzed at 30 days post-induction.’

And in the current submission, we discuss this dichotomy (page 12):

‘Since CNTF by itself does not generate a long-term neuroprotective effect in WT RGCs, but combining CNTF with SARM1(-) inactivation enhances RGC survival beyond that of SARM1(-)’s effect alone, this suggests that the combination of CNTF + SARM1(-) provides a synergistic neuroprotective effect.’

The topic of whether combined CNTF + SARM1(-) treatment fails to improve RGC survival at even more extended times (>60 days), and improves visual function (using LE-pigmented animals to maximize visual acuity) in addition to enhancing survival is the topic for another study we hope to perform. We add in the discussion:

‘Future experiments will determine whether this combination both preserves visual function (via acuity and electrophysiologically) and provides even longer term (60d or longer) RGC survival.’

  1. Oligodendrocyte Preservation (Lines 435-436): The claim of higher oligodendrocyte preservation lacks supporting data in the presented experiments, as myelin preservation is not exclusively dependent on oligodendrocyte viability. To support this assertion, specific staining for oligodendrocytes would be recommended (Oligo2...).

We have changed the statement in results section 3.4 to:

‘suggesting that reducing- or eliminating SARM1 expression in oligodendrocytes also may improve oligodendrocyte preservation.’

And in the discussion (4.1) we add:

‘However, it is important to note that the simple presence of MBP does not by itself delineate improved oligodendrocyte survival. Future experiments quantifying oligodendrocytes and OPCs using SOX10 will help enable evaluation of this question.’

Minor Edits and Typos:

Lines 341-345: Considering that the reference pertains to the authors, it is advisable to state: "In our previous studies..."

In our 12/18/23 submission, this corresponds to line 31. This has been changed to:

‘Our previously reported study, utilizing similar induction conditions showed mean values ranging from 71.4±2.4% sem RGC loss when the ≥500um threshold effect was considered[31], to 53.2±7.1% sem RGC loss when all animals (subthreshold (<500um) as well as threshold≥500um) are included.’

Lines 165-167: Please review this sentence; an open parenthesis is present without a corresponding closed parenthesis.

Corrected. (INL)-INL to (INL-INL)

Lines 338-339: Please review this sentence; an open parenthesis is present without a corresponding closed parenthesis.

Corrected.

Line 223: Add the term ‘Fig’ to “(1G...”

Added.

References that can be considered.    

Lines 174-175: Regarding Brn3a as a reliable marker for RGCs, a reference could help: 19264888, 36594396.

Added: Nadal-Nicolas et al (2009) IOVS 50 (Now ref [32]). Thank you!

Line 640: a study focused on axonal damage in RGCs has demonstrated the protective effect of BDNF in rats: 21315070.

Included: Sanchez-Migallon (2011) EER (92) (Now ref [40]).